# Recent Advances of Calcium Carbonate Nanoparticles for Biomedical Applications

**DOI:** 10.3390/bioengineering9110691

**Published:** 2022-11-15

**Authors:** Pengxuan Zhao, Yu Tian, Jia You, Xin Hu, Yani Liu

**Affiliations:** 1Department of Medical Ultrasound, Tongji Hospital, Tongji Medical College, Huazhong University of Science and Technology, Wuhan 430030, China; 2Jiangsu Hengrui Pharmaceuticals Co., Ltd., Shanghai 200000, China; 3School of Pharmacy, Tongji Medical College, Huazhong University of Science and Technology, Wuhan 430030, China

**Keywords:** CaCO_3_ NPs, drug delivery, diagnosis, theranostics

## Abstract

Calcium carbonate nanoparticles have been widely used in biomedicine due to their biocompatibility and biodegradability. Recently, calcium carbonate nanoparticles are largely integrated with imaging contrast and therapeutic agents for various imaging and therapeutic approaches. In this review, we first described the advantages and preparation methods of calcium carbonate nanoparticles, then the state-of-the-art progress of calcium carbonate nanoparticles in diagnosis, treatment and theranostics was summarized. Finally, we discussed the challenges and recommendations for future studies of the calcium carbonate nanoparticles.

## 1. Introduction

The rapid development of nanotechnology over the past few decades has led to the approval of multiple nanoparticle (NP)-based drug delivery systems in clinics [1,2]. Furthermore, a large number of NPs are undergoing clinical trials or preclinical studies [3]. These NPs can be roughly divided into organic NPs and inorganic NPs. Among different inorganic materials, calcium carbonate (CaCO_3_) NPs have gained much attention due to their excellent biocompatibility and biodegradability, as well as easy preparation and pH sensitivity [4]. CaCO_3_ exists as an amorphous calcium carbonate (ACC) phase, two hydrated metastable phases (calcium carbonate hexahydrate and monohydrocalcite), and three anhydrous crystalline polymorphs (calcite, aragonite, vaterite) [5]. Among them, ACC phase displays the highest solubility and is the precursor of anhydrous crystalline polymorphs, which is easily crystallized in solutions to form polymorphs [6].

By the combination of CaCO_3_ NPs with imaging contrast agents, different imaging modalities such as fluorescence imaging (FLI), magnetic resonance imaging (MRI) and ultrasound (US) imaging could be realized. By the combination of CaCO_3_ NPs with drugs, diverse treatments including chemical therapy, gene therapy, photothermal therapy (PTT)/photodynamic therapy (PDT) and immunotherapy could be achieved. Furthermore, by the combination of CaCO_3_ NPs with both contrast agents and drugs, multimodal theranostics could be reached. Therefore, the development of CaCO_3_ NPs would contribute to the diagnosis, treatment and theranostics of diseases.

In this review, we will first summarize the advantages and preparation methods of CaCO_3_ NPs. Then, CaCO_3_ NP-based biomedical applications will be classified in detail. Finally, we will discuss the challenges and recommendations for future studies of CaCO_3_ NPs.

## 2. The Advantages of CaCO_3_ NPs

### 2.1. Excellent Biocompatibility/Biodegradability and pH-Sensitive Property

In biological systems, calcium carbonate and calcium phosphate are important components of bones, shells or teeth [7]. Therefore, it is believed that CaCO_3_-based drug delivery systems have excellent biocompatibility due to their chemical similarity with tissues. Furthermore, some common NPs such as Au, Ag, Se, Cr, TiO_2_ and ZnO have been demonstrated to improve mutation frequency and reactive oxygen species production, thus, leading to cell apoptosis [8,9]. In contrast, CaCO_3_ NPs are one of the safest biomaterials because their by-products (only Ca^2+^ and CO_3_^2−^) already exist in the blood.

In addition, CaCO_3_ NPs are stable under normal blood pH (7.4) while decompose quickly in an acidic tumor microenvironment, of which facilitates tumor-targeted delivery [10].

### 2.2. Ease of Preparation and Surface Modification

The preparation of CaCO_3_ NPs only needs common salts without organic solvents in most cases, which makes them low-cost [11]. Moreover, the surface of CaCO_3_ NPs can be modified with targeted moiety, which promotes these CaCO_3_ NPs to arrive at the target sites [12].

## 3. The Preparation Methods and Controlled Release of CaCO_3_ NPs

So far, the commonly used preparation methods of CaCO_3_ NPs include the precipitation method [13], gas diffusion [14], flame synthesis [15], decomposition of cockle shells [16], biomineralization and so on [17,18]. Among them, solution precipitation, microemulsion and gas diffusion methods have been widely used for CaCO_3_ NP-based drug delivery systems.

### 3.1. Solution Precipitation Method

The solution precipitation method is the most established technique for CaCO_3_ NP preparation, which uses the reaction between the Ca^2+^ and CO_3_^2−^ aqueous solution. This method could produce large quantities of CaCO_3_ NPs without a surfactant, thus, reducing the production cost. Because of the mild preparation conditions, many bioactive species, including small molecule drugs, genes and proteins, could load into CaCO_3_ NPs during the precipitation process [4]. Notably, the synthesis parameters such as pH, temperature, ion concentration, stirring speed, solvent species and additives are often used to control the size, shape and phase of CaCO_3_ NPs [13].

### 3.2. Microemulsion Method

As an extension of the precipitation method, the microemulsion methods are widely used for CaCO_3_ NP preparation and gene encapsulation [19,20]. Microemulsion methods contain the reversed microemulsion (water in oil, W/O) method and double emulsion method. The reversed microemulsion method used the W/O microemulsion droplets as the nano-reactors [19]. First, “calcium microemulsion” and “carbonate microemulsion” were, respectively, prepared through adding the Ca^2+^ or CO_3_^2−^ aqueous phase into an organic phase. Then, “calcium microemulsion” and “carbonate microemulsion” were mixed to form CaCO_3_ NPs. Finally, a centrifuge was used to separate the CaCO_3_ NPs. For example, Huang et al. developed CaCO_3_ NP loading with the therapeutic peptide by the reversed microemulsion method for lung cancer treatment [21].

The double emulsion method is similar to the reversed microemulsion method [20]. Firstly, W/O “calcium microemulsion” was prepared the same as the reversed microemulsion method. Then, a great deal of aqueous phase (consisting of CO_3_^2−^) was mixed with “calcium microemulsion” to form the W/O/W double emulsion. CaCO_3_ NPs were formed through the Ca^2+^ and CO_3_^2−^ reaction in the W/O/W double emulsion.

In general, through the microemulsion method, the structure, size and crystallinity of CaCO_3_ NPs could be regulated by optimizing the surfactants, temperature, pH and ion concentration [22].

### 3.3. Gas Diffusion Method

The gas diffusion method is mainly used for preparing ACC loading with small molecule drugs [14]. As shown in Figure 1, CaCl_2_ was dissolved in ethanol and transferred into a glass bottle. Then, the bottle was left in a desiccator along with another bottle of ammonia bicarbonate. CO_2_ and NH_3_ were generated from ammonium bicarbonate, then dissolved in the ethanol solution to form CO_3_^2−^ and NH_4_^+^. Under an alkaline condition caused by NH_4_^+^, CO_3_^2−^ was reacted with Ca^2+^ to form ACC. In this method, the size, shape and polymorph of the prepared ACC could be controlled through changing the additives, temperature and Ca^2+^ concentration [23].

### 3.4. Controlled Release of CaCO_3_ NPs

CaCO_3_ NPs could improve the pharmacokinetics of loading drugs through a controlled release, thus, reducing the side effects and enhancing the treatment effect. CaCO_3_ NPs release the drugs by three ways, including diffusion, carrier dissolution and recrystallization [24]. pH is the key parameter for the controlled release of CaCO_3_ NPs. Under acidic conditions, free protons react with CO_3_^2−^ to form HCO_3_^−^, then dissolve CaCO_3_ NPs and accelerate the release of loading drugs [25].

## 4. The Biomedical Applications of CaCO_3_ NPs

### 4.1. CaCO_3_ NPs for Diagnosis

Through combining CaCO_3_ NPs with fluorophores or paramagnetic elements (such as Mn^2+^, Gd^3+^), FLI and MRI could be realized [26,27]. Moreover, CaCO_3_ NPs themselves can produce CO_2_ bubbles under acidic conditions, which can then enhance the US imaging signal. For example, Kim et al. prepared CaCO_3_ NPs for US imaging [28]. After an intravenous injection, the prepared CaCO_3_ NPs showed a remarkable US contrast enhancement in the tumor tissue. In addition, Yi and co-workers reported membrane-cloaking nanoconjugates comprising NaGdF_4_ and CaCO_3_ NPs [27], which displayed more than a 60-fold contrast enhancement compared with Magnevist (commercially used contrast agent) in tumor MRI (Figure 2).

### 4.2. CaCO_3_ NPs for Treatment

Because of the excellent biocompatibility/biodegradability, pH-sensitive property, ease of preparation and surface modification, CaCO_3_ NPs have been widely used as carriers for a variety of treatments including chemical therapy [29], gene therapy [21], PTT/PDT [30] and combination therapy [31]. Moreover, CaCO_3_ NPs themselves could be used as Ca^2+^ generators which induce immunogenic cell death (ICD) and autophagy to activate immunotherapy [12].

#### 4.2.1. CaCO3 NPs as Carriers for Chemical Therapy

CaCO_3_ NPs were able to load both hydrophobic and hydrophilic molecules, making them suitable carriers for chemotherapy [9]. For instance, Wang et al. designed monostearin-coated CaCO_3_ NPs for doxorubicin (DOX) loading [29]. Monostearin coating induced a lipase-triggered DOX release in a lipase-overexpressed tumor site, which improved the drug penetration (Figure 3).

#### 4.2.2. CaCO_3_ NPs as Carriers for Gene Therapy

Gene therapy works by substituting or silencing the defective gene to achieve the therapeutic effect [32]. However, it has been a challenge for nucleic acid delivery due to their negative charge, large size and easy degradation [33]. CaCO_3_ NPs could bind with nucleic acids, making them promising vehicles for gene therapy [21]. For example, He et al. constructed CaCO_3_ NPs for vascular endothelial growth factor small interfering RNA (VEGF siRNA) delivery [34]. Both in vitro and in vivo results demonstrated that CaCO_3_ NPs are a suitable system for siRNA delivery. In another study, Chen et al. synthesized CaCO_3_ NPs and modified them with polyethyleneimine (PEI), named as PEI-CaCO_3_ NPs, which could be used for p53 gene adsorption [35]. After transfected, p53-loaded PEI-CaCO_3_ NPs significantly decreased the proliferation of tumor cells.

#### 4.2.3. CaCO_3_ NPs as Carriers for PTT/PDT

PTT and PDT have become promising strategies for cancer therapy because of the noninvasiveness and specific selectivity [36,37]. Recently, Xue et al. fabricated a nanocomposite consisting of CaCO_3_, indocyanine green (ICG) and polydopamine (PDA), named as Fe_3_O_4_@PDA@CaCO_3_/ICG (FPCI) NPs, which can achieve the combination of PDA-based PTT and ICG-based PDT (Figure 4) [30].

#### 4.2.4. CaCO_3_ NPs as Ca^2+^ Generators for Immunotherapy

Immunotherapy works by activating the immune system for searching and destroying cancer cells [38]. CaCO_3_ NPs can be used not only as carriers for immunotherapy drugs themselves, but also could increase Ca^2+^ concentration, thus, inducing immunogenic cell death (ICD) and autophagy [39,40]. Most recently, Zheng et al. prepared polyethylene glycol (PEG)-decorated CaCO_3_ NP loading with curcumin (namely, ^PEG^CaCUR) [39]. ^PEG^CaCUR NPs can serve as a Ca^2+^ nanomodulator to induce Ca^2+^ overload, thus, enhancing the ICD effect and eventually inhibiting tumor growth and migration (Figure 5a). In another study, An et al. designed ovalbumin (OVA)-loaded CaCO_3_ (OVA@CaCO_3_) NPs as a Ca^2+^ nanogenerator to destroy the autophagy inhibition condition in dendritic cells, promote the damage-associated molecular patterns (DAMPs) and release and upregulate the pH of the tumor microenvironment (Figure 5b) [40].

#### 4.2.5. CaCO_3_ NP-Based Combination Therapy

Combination therapy is able to notably decrease multidrug resistance and increase efficiency [41]. CaCO_3_ NPs are commonly used for the co-delivery of chemotherapeutics and gene drugs, which realized the combination of chemotherapy and gene therapy [31]. For example, Xiang’s group designed a lipid-coated CaCO_3_ NPs for the co-delivery of sorafenib and miR-375 (miR-375/Sf-LCC NPs, Figure 6a) [42]. Both in vitro and in vivo results proved that miR-375/Sf-LCC NPs are promising carriers for combination therapy. In another study, Kong et al. developed gold nanorods@CaCO_3_ NPs coated with dextran and phospholipid for the incorporation of different molecules, including DOX, 17-(allylamino)-17-demethoxygeldanamycin, afatinib and amylase (Figure 6b) [43]. This platform has great potential for the combination of PTT and chemotherapy.

### 4.3. CaCO_3_ NPs for Theranostic

The therapeutic effect could be significantly improved through the rational design of novel theranostic platforms with both imaging and treatment functions [44]. CaCO_3_ NPs have shown potential in both diagnosis and therapy, which encourages researchers to design theranostic CaCO_3_ NPs for achieving imaging-guided treatment [4]. Specifically, CaCO_3_ NP-based theranostic platforms can be classified as three types according to the imaging mode, including US imaging-guided therapy [45], FLI-guided therapy [46] and MRI-guided therapy [47].

#### 4.3.1. US Imaging-Guided Therapy

CaCO_3_ NPs can generate CO_2_ bubbles and display potential as a US contrast agent in the acidic tumor microenvironment. As a typical paradigm, Min et al. developed DOX-loaded CaCO_3_ NPs that express US imaging and chemotherapy for cancer theranostics (Figure 7a) [45]. These NPs displayed a strong echogenic signal and long echo persistence, as well as a simultaneous DOX release at the tumor site, which exhibited efficient antitumor effects guided by US imaging. Recently, Feng and co-workers reported CaCO_3_ NP loading with hematoporphyrin monomethyl ether (HMME, a sonosensitizer) [48]. Under US irradiation, generated CO_2_ bubbles could lead to cavitation-mediated necrosis and be used as US contrast agents. Meanwhile, HMME can produce reactive oxygen species for sonodynamic therapy (Figure 7b). These nanoplatforms provided the US imaging-guided cavitation/sonodynamic combined therapy, which highlighted the possibility of cancer theranostics.

#### 4.3.2. FLI-Guided Therapy

CaCO_3_ NPs could be constructed as FLI-guided therapy nanoplatforms through the co-delivery of FLI contrast and therapeutic agents. For example, Huang et al. designed a theranostic CaCO_3_ NP encapsulation with DOX and fluorescence contrast agent indocyanine green (ICG) for chemotherapy and fluorescence/US dual-mode imaging [46]. The prepared CaCO_3_ NPs showed a satisfactory treatment effect guided by dual-mode imaging, which demonstrated a promising strategy for dual-mode theranostics.

#### 4.3.3. MRI-Guided Therapy

CaCO_3_ NPs can also load with MRI contrast and therapeutic agents for realizing MRI-guided therapy. For instance, Gorin’s group prepared a CaCO_3_ NP-capsuling paramagnetic element (Fe_3_O_4_) and DOX, which could be used for an MRI/photoacoustic imaging-guided precise drug release [47].

## 5. The Challenges and Recommendations for Future Studies of CaCO_3_ NPs

Although CaCO_3_ NPs have been widely investigated for diverse biomedical applications including diagnosis, treatment and theranostics due to their excellent biocompatibility/biodegradability and pH-sensitive property, as well as their ease of preparation and modifications, there are still several challenges that need to be addressed for clinical translation.

First, long-term potential risks of CaCO_3_ NPs need to be noticed [49,50]. Although calcium is an essential element in humans, overloaded calcium could induce thrombosis, hypercalcemia and other potential dangers [51]. Furthermore, the most recent studies only evaluated the short-term toxicity of mice through describing organ damage and immune responses after a CaCO_3_ NP injection, which was obviously insufficient for a biosafety evaluation. Thus, for the clinical translation of CaCO_3_ NPs, it is necessary to systematically assess the long-term effects of CaCO_3_ NPs from rodent models to mammalian models [4].

Second, the present preparation processes of CaCO_3_ NPs are instable, which easily leads to large particles [9]. Thus, it is necessary to design precise methods for size control, components, and surface modifications, sequentially achieving a large-scale production of CaCO_3_ NPs.

Third, the drug release kinetics from CaCO_3_ NPs is difficult to predict. Although the pH-sensitive property of CaCO_3_ NPs has been widely studied, their release under a normal pH has not been evaluated in detail.

## 6. Conclusions

In summary, CaCO_3_ NPs have great potential in biomedical applications due to their excellent properties, such as biocompatibility/biodegradability, pH-sensitivity, ease of preparation and surface modifications. Although much has been carried out, more efforts are still needed to solve the above challenges. We believe that more efficient CaCO_3_ NPs will be developed as safe carriers for the diagnosis, treatment and theranostics of diseases.

## Figures and Tables

**Figure 1 bioengineering-09-00691-f001:**
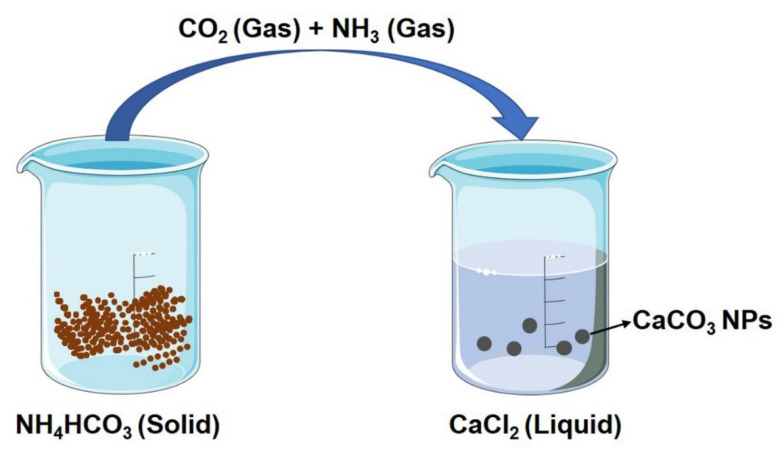
Illustration of the gas diffusion method. CO_2_ and NH_3_ were generated from ammonium bicarbonate, which then dissolved in the ethanol solution and reacted with Ca^2+^ to form CaCO_3_ NPs.

**Figure 2 bioengineering-09-00691-f002:**
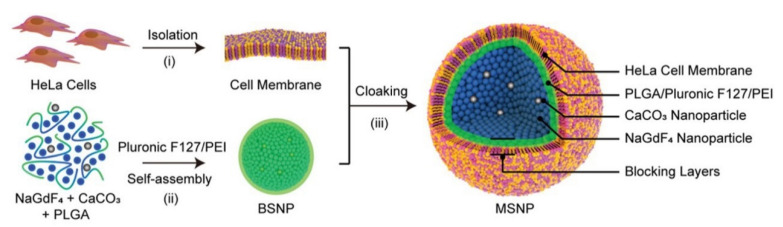
Schematic illustration of the prepared CaCO_3_ NPs for MRI. Reprinted with permission. Copyright 2019, Wiley-VCH [27].

**Figure 3 bioengineering-09-00691-f003:**
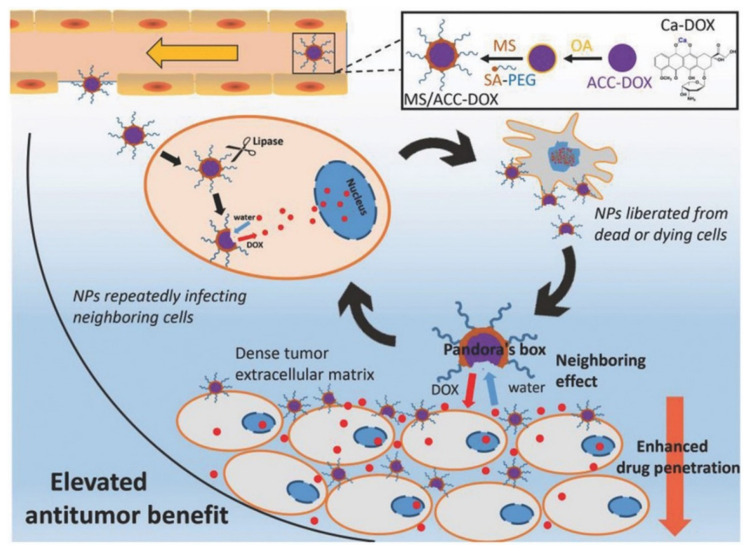
Illustration of the formation and elevation of the designed CaCO_3_ NPs. Reprinted with permission. Copyright 2018, Wiley-VCH [29].

**Figure 4 bioengineering-09-00691-f004:**
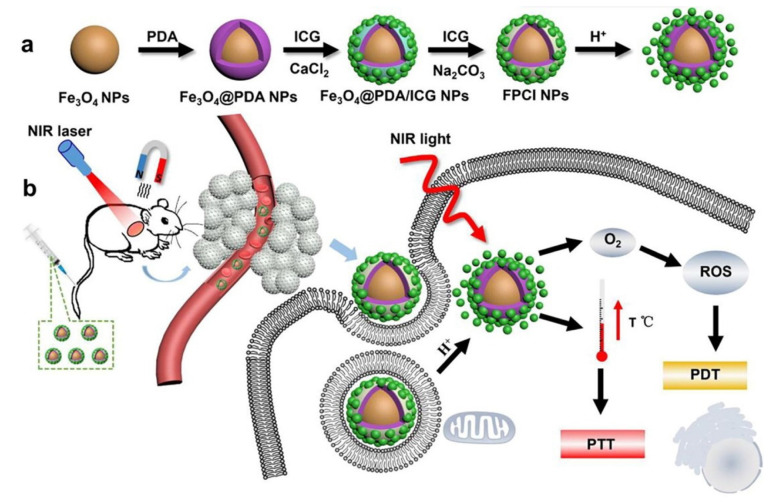
Schematic of the preparation process (**a**) and PTT/PDT treatment (**b**) of FPCI NPs. Reprinted with permission. Copyright 2018, Elsevier [30].

**Figure 5 bioengineering-09-00691-f005:**
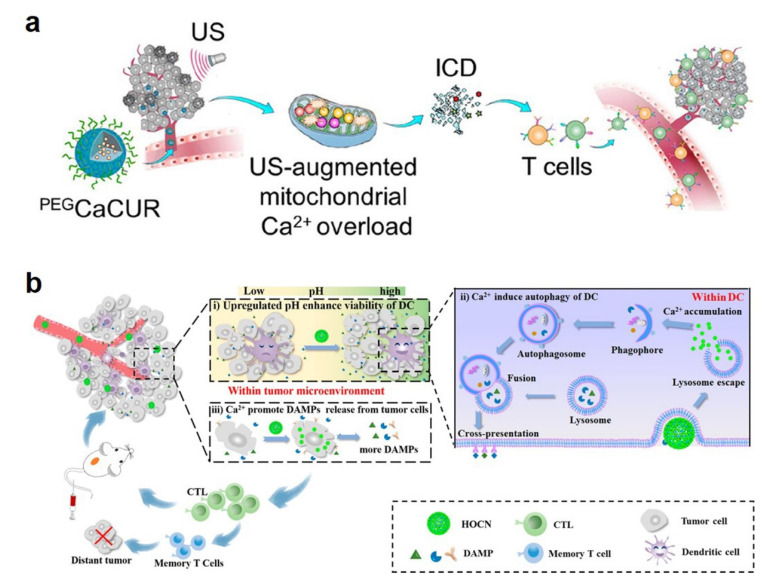
(**a**) Schematic illustration of ^PEG^CaCUR NPs-based immunotherapy. Reprinted with permission. Copyright 2021, American Chemical Society [39]. (**b**) Schematic diagram of OVA@CaCO_3_ NP-mediated immunotherapy. Reprinted with permission. Copyright 2020, American Chemical Society [40].

**Figure 6 bioengineering-09-00691-f006:**
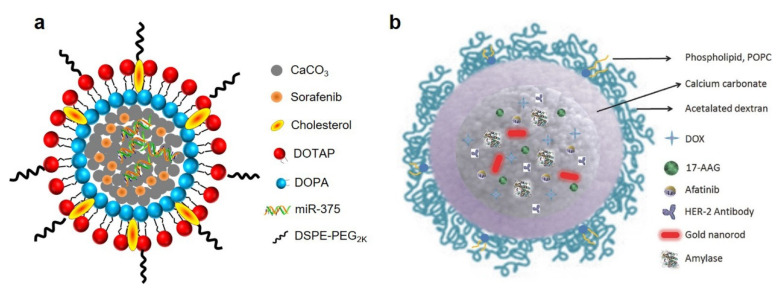
(**a**) Schematic representation of miR-375/Sf-LCC NPs. Reprinted with permission. Copyright 2018, Elsevier [42]. (**b**) Schematic representation of gold nanorods@CaCO_3_ NPs. Reprinted with permission. Copyright 2016, Wiley-VCH [43].

**Figure 7 bioengineering-09-00691-f007:**
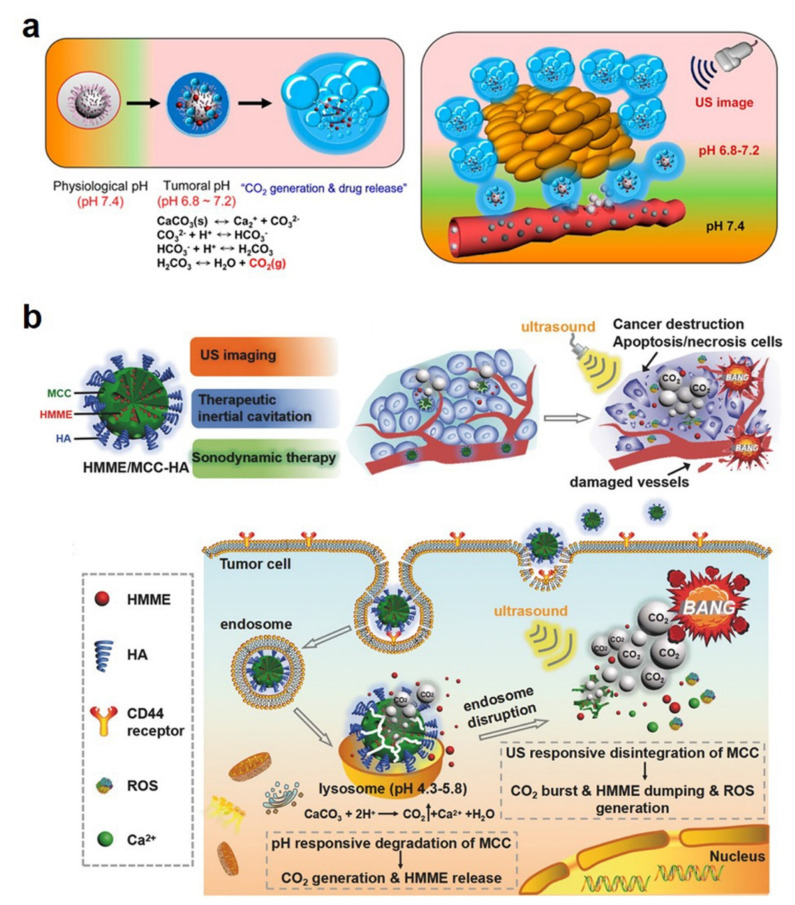
(**a**) Bubble generation and drug release of the designed CaCO_3_ NPs. Reprinted with permission. Copyright 2015, American Chemical Society [45]. (**b**) Schematic illustration of the US imaging-guided cavitation/sonodynamic combined therapy. Reprinted with permission. Copyright 2017, Wiley-VCH [48].

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
