# Peer review of "Recent Advances of Calcium Carbonate Nanoparticles for Biomedical Applications"

_bioengineering, 2022, doi:10.3390/bioengineering9110691_

Round 1
Reviewer 1 Report
Dear Authors;
Re: [bioengineering-2006601
Title: "Recent advances of calcium carbonate nanoparticles for biomedical applications"]
Your manuscript, a Review article, describes advantages of calcium carbonate nanoparticles applied for drug delivery and diagnosis (theranostic use). You have also mentioned two of the available methods for their manufacture.
Please find my comments / suggestions below:
1. Numerous English language mistakes need to be corrected. Examples include:
"... we first described in detail with the advantages and preparation methods ..." in the Abstract;
" ... thus leading cell apoptosis ..." Line 53 and 54; SECTION 2.1.;
"... are already existed in the blood ..." Line 55;
"... which facilitating tumor targeted drug delivery ..." Line 57;
2. Why there is no gap between the last words proceeding [References]?
Example: "... clinic[1,2] ..." In the Introduction section;
3. Please add "and" before the last item in a list.
Example:
"... and three anhydrous crystalline polymorphs (calcite, aragonite, vaterite) [5].";
4. Section 3. The preparation methods of CaCO3 NPs
only contains 2 sub-sections. Please add more methods to this section;
5. Figure 1 legend requires more details;
6. Please check Figure 2 legend "... diagnosed CaCO3 NPs ..." needs correction;
7. Please correct the sentence in Section 4.2. It should be: " ... for a variety of treatments ...". I suggest you expand this section by adding more sentences;
8. Section 4.3. needs to be expanded and also needs References;
9. You mentioned: "The rapid development of nanotechnology over the past few decades has led to the approve of multiple nanoparticles ...". Is there any approved calcium carbonate-based nanoparticles for Human use? A Table listing approved NPs will be useful to the readers;
10. Please correct the sentence "So, it is necessary to design precision preparation ..." in Lines 228 - 230;
11. Conclusion section is missing. I also suggest "Recommendations For Future Studies" be added.
Reviewer 2 Report
The manuscript by P. Zhao et al. entitled “Recent advances of calcium carbonate nanoparticles for bio-medical applications” is a short actual review of published papers about the preparations methods and applications in medicine of those nanoparticles. The advantages of these nanoparticles are given. A critical conclusion showing the challenges and missing studies in the application of these nanoparticles completes the manuscript.
The manuscript is straightforward written. The reader can follow it easily and gains a quick overview on the literature.
After checking by a native English speaker it can be published. I like already to mention a few of the language corrections necessary:
• Line 53-54: .... thus leading to cell .....
• Line 55: ... are already existing in ....
• Line 70: The co-precipitation ....
• Line 71: ... which uses reaction ...
• Line 74: ... of the co-precipitation method, microemulsion methods are ....
• Line 81: Finally, a centrifuge was ...
• Line 86: The double emulsion ....
• Line 87: ... was prepared in the same way as ....
• Line 92: The gas diffusion method ...
• Line 99: Illustration of the gas diffusion method
• Line 117: .... PTT/PDT ....
• Line 136: ... and modified them with poly
• Line 150: ... NPs can be used not only as ...
• Line 221:calcium is an essential element ....
• Line 225: this sentence seems to be incomplete
• Line 227:.. NPs are instable, which ....
Reviewer 3 Report
The manuscript "Recent advances of calcium carbonate nanoparticles for biomedical applications" by Pengxuan Zhao is a short review focused on selected applied aspects of calcium carbonate nanoparticles in biomedicine. Despite this subject is very topical and appropriate for the Bioengineering, the manuscript in its present form requires many essential improvements. Sections 2 and 3 are too short and thus they do not provide sufficient detail on the contemporary status in the field. The section 4 is indeed interesting and useful but it also gives a very limited number of illustrative examples on biomedical uses of composite systems encompassing CaCO3 NPs as one of the componentns. In most cases, the role played by calcium carbonate itself is not quite evident.
Section 3 should be significantly extended to emphasize capabilities to control size, shape, crystallinity and phase composition of CaCO3 nanoparticles during their synthesis.
Also, preparation methods to load/modify CaCO3 nanoparticles with bioactive species should be surveyed as well as methods for the controlled release thereof. Probably, a separate dedicated section would be needed to outline this information.
Probably, it would make sense to divide the current section 4 into two subsections with specific examples of applications, where CaCO3 plays a crucial role and by itself or it is simply used as an inert biocompatibe carrier. In each case, it would make sense to formulate the most important factor in favor of using CaCO3 rather than any other system.
Line 52. The phrase should be reformulated since Se, TiO2, ZnO are not metallic.
The term co-precipitation (3.1 section title) is typically applied to a simultaneous precipitation of two or more components, which is not the case in the manuscript's context. The authors rather describe a simple precipitation.
There are numerous acronyms throughout the text left without explanation, e.g., FLI, MRI, US, PTT, PDT. Please, correct.
Round 2
Reviewer 1 Report
Dear Authors;
Re: [bioengineering-2006601
Title: "Recent advances of calcium carbonate nanoparticles for biomedical applications"]
Thanks for addressing suggestions and comments.
Reviewer 3 Report
The revisions made by the authors are appropriate. The manuscript in its present form can be recommended for acceptance.